# Molecular Profile of Canine Hemangiosarcoma and Potential Novel Therapeutic Targets

**DOI:** 10.3390/vetsci10060387

**Published:** 2023-06-05

**Authors:** Pedro Antônio Bronhara Pimentel, Antonio Giuliano, Paweł Marek Bęczkowski, Rodrigo Dos Santos Horta

**Affiliations:** 1Department of Veterinary Clinic and Surgery, Federal University of Minas Gerais, Belo Horizonte 31270-901, MG, Brazil; rodrigohvet@gmail.com; 2Department of Veterinary Clinical Science, Jockey Club College of Veterinary Medicine, City University of Hong Kong, Hong Kong, China; agiulian@cityu.edu.hk (A.G.); pbeczkow@cityu.edu.hk (P.M.B.); 3Veterinary Medical Centre, City University of Hong Kong, Hong Kong, China

**Keywords:** angiosarcoma, dog, NRAS, PIK3CA, VEGF, CDKN2A, PTEN, TP53

## Abstract

**Simple Summary:**

Hemangiosarcoma (HSA) is a highly aggressive vascular tumor. It is the most common splenic cancer and the cause of non-traumatic abdominal hemorrhage in dogs. The short overall survival and high dissemination potential of HSA demonstrate the necessity of new and more effective therapies, especially with specific tumoral targets. This study investigates recent advances in molecular aspects of canine hemangiosarcoma and presents promising therapeutic targets.

**Abstract:**

Canine hemangiosarcoma (HSA) is a relatively common neoplasia, occurring mainly in the skin, spleen, liver and right atrium. Despite the numerous studies investigating the treatment of canine HSA, no significant improvement in survival has been achieved in the last 20 years. Advancements in genetic and molecular profiling presented molecular similarities between canine HSA and human angiosarcoma. It could therefore serve as a valuable model for investigating new and more effective treatments in people and dogs. The most common genetic abnormalities in canine HSA have been found in the phosphatidylinositol-4,5-bisphosphate 3-kinase catalytic subunit alpha (PIK3CA) and neuroblastoma RAS viral oncogene homolog (NRAS) pathways. Mutations are also found in tumor protein p53 (TP53), phosphatase and tensin homolog (PTEN) and cyclin dependent kinase inhibitor 2A (CDKN2A). Known abnormal protein expression could be exploited to trial new target treatments that could be beneficial for both canine and human patients. Despite the high expression of vascular endothelial growth factor (VEGF) and its receptor (VEGFR), no correlation with overall survival time has ever been found. In this review, we explore the most recent developments in molecular profiling in canine HSA and discuss their possible applications in the prognosis and treatment of this fatal disease.

## 1. Introduction

Angiosarcoma is an aggressive cancer that carries a poor prognosis in people, with 5-year survival tallying at less than 30% [1]. In dogs, malignancies of lymphatic vessels are rarely described, while a tumor of the blood vessels—hemangiosarcoma (HSA)—is relatively common. HSA may arise in any region of the body, especially the skin, spleen, liver and atrium [2,3]. Skin lesions can be multiple, often small and related to ultraviolet radiation (UV) and actinic damage, especially in the lower abdomen of dogs with white or thin hair coats [4,5]. However, HSA may also occur in the subcutis, muscle and visceral organs. Non-cutaneous HSAs are highly aggressive tumors with a high metastatic rate [6,7,8,9]. The splenic form shows nonspecific clinical signs during its initial progression, yet is the most common cause of acute hemoperitoneum [10,11] with a high rate of metastases to the liver, mesentery, abdominal lymph nodes and lungs [7,8]. Cardiac HSA has a poor prognosis due to its fast local growth and progression, with frequent concurrent involvement of the lungs, spleen and liver [6]. HSA has a variable breed predisposition. Whereas dermal neoplasia seems to be directly caused by UV exposure in white dog breeds, especially in Pitbull Terrier and Boxer [4,5], the visceral forms are overrepresented in breeds such as Golden Retriever, Labrador Retriever, German Shepherd and Miniature Schnauzer [3,7,8,12,13,14].

The anatomical form of HSA is directly related to its cytogenetic origin. Older studies suggest the development of HSA from transformed mature endothelial cells based on the histological presentation and negative staining for leukocytes and histiocytes but positive for CD31, CD105, CD146, VEGFR, factor VIII and avb3-i-integrin (from activated endothelial cells) [15,16]. This theory may be true for cutaneous HSA, which justifies the biological behavior and rare identification of metastases. However, this classic model, in which a primary tumor arises in an organ and metastasizes from there, does not completely explain the behavior of canine HSA in its other forms [15,16,17].

A second theory has been gaining momentum over the last 15–20 years. Except for the actinic cutaneous form, canine HSA could arise from a precursor (pluripotent) endothelial cell. This theory has been confirmed in several molecular studies, based on the expression of CD34, CD45, CD133 and KITr, which are endothelial precursor cell proteins [15,16]. These cells can be identified in the circulation, with values higher than 0.5% in dogs with HSA and less than 0.3% in dogs without HSA or dogs with HSA already submitted to surgery. These pluripotent cells leave the bone marrow to disseminate to different parts of the body [15]. Their survival, growth and proliferation are dependent on the microenvironment. Possibly, the spleen is a more favorable environment, followed by the liver and right atrium. One study classified HSA according to distinct endothelial, myeloid and hematopoietic markers (CD14, CD34, D105, CD115, CD117, CD133 and CD146), suggesting that it could arise from different pluripotent progenitors [18], further classified as angiogenic, inflammatory and adipogenic [16]. Figure 1 illustrates an adaptation of the hematopoietic progenitor theory for HSA dissemination presented by Kim et al. (2015) and Lamerato-Kozicki et al. (2006) [15,16].

The diagnosis of canine HSA is commonly achieved by histopathology due to the inherent limitations of cytology [21,22]. Histopathology, however, is not always possible without performing invasive surgery. A novel minimally invasive diagnostic test (liquid biopsy of a blood sample) was recently validated. Although based on a small sample size (*n* = 12), this test was shown to achieve a specificity of 83.3% for the diagnosis of canine HSA [23]. A different liquid biopsy test, which measured plasma nucleosome concentrations for canine HSA, included a six times higher number of cases than the previous study (*n* = 77), demonstrated improved specificity (97%) and exhibited a sensitivity of 81.2% [24]. However, the sensitivity of liquid biopsy is likely to be affected by the stage of the disease, with lower stages less likely to be detected [23,24]. Even though more research is needed, liquid biopsy could become a non-invasive screening exam, especially for certain breeds of dogs with a high predisposition for HSA.

The prognosis for actinic cutaneous lesions is favorable and surgery is usually curative with long survival rates [25,26]. Nevertheless, aggressive systemic forms of disease are associated with poor prognosis with a median survival time (MST) of 23–292 days [4,25,26,27].

Despite the numerous studies investigating the treatment of canine HSA, no significant improvement in survival has been achieved in the past 20 years. Standard treatment consists of surgical resection followed by anthracycline-based chemotherapy at the highest tolerated dose. The most used tyrosine kinase inhibitors in dogs, toceranib and masitinib, were not effective in the treatment of canine HSA [28,29]. In a study of dogs with splenic hemangiosarcoma treated with standard splenectomy and subsequent adjuvant doxorubicin, the addition of toceranib phosphate did not result in an improvement in overall survival or disease-free interval [29]. Although masitinib has some anti-proliferative effects on canine HSA cells in vitro, the in vivo effect has not been reliably demonstrated in clinical trials [28]. The addition of maintenance metronomic chemotherapy was not effective in enhancing overall survival either [30]. However, treatment with metronomic chemotherapy with or without thalidomide could be as effective as the standard-of-care adjuvant doxorubicin [31,32,33,34]. The outcomes of studies exploring different treatments, including immunotherapy strategies, for HSA have been largely disappointing [35,36,37,38,39]. Advancements in the genetic and molecular profiling of specific cancers have opened new avenues for personalized and targeted treatment in both human and veterinary medicine. Few studies have investigated the molecular profiling of HSA in dogs [40,41,42]. Although the morphological and immunohistochemical studies of canine HSA have deciphered important aspects concerning its origin and behavior, research on the molecular basis of the disease may be more important for targeted therapy. Several molecular similarities between canine HSA and human angiosarcoma could be advantageous for the development of targeted therapies [43,44].

Histone acetylation is an epigenetic mechanism of cancer modulation, recently studied for potential targeted therapies [45,46]. One in vitro study has demonstrated its potential for canine HSA: histone acetylation levels were high in canine HSA cell lines and some in vivo cases, suppressed by a bromodomain and extraterminal domain inhibitor (BETi), JQ1 [46]. The potential for this therapy is still unclear, so new studies are required to characterize it properly.

A study analyzed DNA copy number variations and found distinct patterns of gain or loss of specific loci in dogs diagnosed with intrabdominal HSA [47]. Significative gains occurred in chromosome 13 in VEGFR2, PDGFRA and KIT genes. In chromosome 12, the dogs also presented VEGFA gene gain, a potential prognostic factor for HSA treatment, especially in Flat-Coated Retrievers, due to the higher rate of gene gain [47,48]. Most cases presented a loss of the CDKN2AIP gene (genomic location 16:49.9), an important tumoral suppressor gene, which encodes p14 and p16 [47]. Data regarding CDKN2AIP mutations in canine HSA are still scarce; however, a recent study revealed mutations in 11% of cases [40], a higher rate compared to mutations of PTEN, which is the most studied tumoral suppressor gene [49,50,51].

Canine HSA has been divided into specific subtypes according to the molecular patterns that might originate from different pathogenic pathways [18,43,44]. The differentiation of three distinct subtypes of HSA) in canines relies on the analysis of specific somatic mutations occurring most frequently in driver genes among affected canine patients: oncogene Phosphatidylinositol-4,5-bisphosphate3-kinase catalytic subunit alpha (PIK3CA), oncogene neuroblastoma rat sarcoma virus (NRAS) activation and tumoral suppressor gene tumor protein p53 (TP53). Changes in the NRAS and PIK3CA pathways may occur in up to 24% (15/50) and 46% (23/50) of canine HSA, respectively, and might be especially interesting for target therapies [18]. Figure 2 presents a summary of the recent literature regarding the main driver mutations associated with canine HSA. Golden retrievers’ HSAs express a different pattern of mutations compared to other pure breeds, present a higher frequency of mutations in AKT and PIK3CA genes and demonstrate the importance of heritable factors in analyzing mutations [41,52]. Thus, they may be beneficial for therapies targeting the products of such genes.

To write this review, a PubMed search was conducted using the following key words: hemangiosarcoma, canine, dogs, angiosarcoma. The search strategy involved evaluating articles spanning 2015 to 2023. Additional studies were incorporated based on their relevance and references from the initially selected articles.

Inclusion criteria considered studies obtained through institutional access or an internet search, especially from peer-reviewed journals. We did not impose any restrictions regarding the characteristics of in vitro and in vivo studies for the selection process. Conventional and well-established treatments for canine HSA, such as surgery and chemotherapies such as doxorubicin and carboplatin, were not extensively evaluated and therefore served as exclusion criteria, considering that the focus of this study was to analyze molecular features. The process entailed several phases, including an initial search, the identification of relevant articles, screening for suitability, an assessment of the eligibility criteria and ultimately determining the final inclusion.

## 2. Hemangiosarcoma Carcinogenesis

Primary canine HSA can occur in distinct organs, at different frequencies. The most commonly diagnosed presentations are splenic, hepatic, cutaneous and cardiac [6,7,54,55]; however, this neoplasm can originate in the lungs, peritoneum, kidneys, skeletal muscles, pleura, oral cavity, pancreas, bones, intestines and virtually every malignant-transformed endothelial vascular tissue [8,9,52,53]. Thus, it should always be considered as a potential differential diagnosis.

The etiopathogenesis of this complex neoplasm relies on genetic predispositions, acquired mutations, hormonal aspects and exposure to environmental carcinogens, such as UV light [4,53,56,57]. These etiological factors can also influence individual clinical presentation, according to the dog’s breed, age, weight and skin characteristics [8,26,52,58]. The impact of gonadal steroids in HSA carcinogenesis is still discussed, with some studies demonstrating an increased risk for the development of HSA in neutered dogs [57,59,60]. Typical mutations in dogs with HSA occur in the TP53 genes, commonly called the “guardian of the genome”, and PIK3CA, active in the PI3K-AKT-mTOR cell proliferation signaling pathway [41,43,44].

Over the last decade, cell culture studies have significantly enhanced our understanding of ’as’s etiopathogenesis and its underlying mechanisms, including the characterization of mutations and cancer mechanisms through genome-wide and transcriptomic analyses [19,20,61,62]. In 2017, Im et al. presented a mechanism of cell migration and invasion through the CXCR4/CXCL12 axis, in which HSA may disseminate [19]. In 2022, Maeda et al. performed the first cloning of canine PIK3CA (GenBank accession no. LC625864), which is highly homologous to the respective human gene. The findings further highlight another potential similarity between human angiosarcoma and canine HSA [61].

A variety of oncogenes and tumoral suppressor genes are associated with the development and progression of HSA in dogs. Table 1 illustrates the genes and their role in canine HSA:

## 3. Mutations and Potential Therapeutic Targets

Table 2 presents the recently studied and potential therapeutic targets, as discussed in the following sections. This table provides details of specific targets and their potential target therapies. The specification of experimental conditions is relevant to determining whether the study was conducted in humans and/or dogs, rather than exclusively focusing on canine HSA.

### 3.1. NRAS

RAS (rat sarcoma) includes a group of small proteins activated by GTP ligation to signal the transduction of diverse signaling pathways related to cell proliferation and survival. The dysregulation of RAS activity can lead to abnormal cell proliferation, with more than 30% of human cancers being driven by activating RAS mutations [79]. The RAS gene group includes HRAS, KRAS and NRAS, that when mutated lead to the carcinogenesis of different neoplasms in humans and domestic animals [80,81,82]. Mutations in NRAS are associated with canine leukemia and HSA [43,80], while changes in KRAS may induce pancreatic and pulmonary carcinomas in dogs [83,84]. HRAS still does not seem to be as important as the other genes in canine cancer pathogenic mechanisms [80]. In canine HSA, mutations in NRAS have been found in 24% (12/50) of HSA cases, independently of anatomical location [43].

The Ras p21 protein activator 1 gene (RASA1) is responsible for the RAS pathway regulation by controlling its ligation with GDP and GTP, and thus is considered a tumor-suppressor gene of the RAS/GAP group. The inhibition of RAS GAPs leads to the upregulation of the Ras pathway, possibly increasing cell proliferation and growth, and ultimately inducing tumorigenesis [85]. RASA1 is crucial for physiological mechanisms such as angiogenesis; however, its mutations are associated with common cancers in humans, including lung, breast, liver and colorectal neoplasms [86,87]. The downregulation of RASA1 may promote tumoral angiogenesis and metastasis [87]. These mutations were identified in 8.5% (4/47) of canine visceral HSA cases [44,64].

Cancers driven by such mutations are difficult to treat because there are no available drugs that can bind to RAS. Nevertheless, the RAF protein, later activated by RAS, can be inhibited by a group of tyrosine kinase inhibitors, including sorafenib, vemurafenib, dabrafenib and regorafenib [88,89].

In people, target treatment with the BRAF inhibitor sorafenib has shown some efficacy [48]. Despite that, there are no published results of clinical trials involving these drugs for canine HSA. Sorafenib has been generally considered quite safe in dogs [72,73,74]. In a small study, sorafenib proved to be safe and effective for dogs with unresectable hepatocellular carcinoma treated with a twice-daily dosage of 5 mg/kg. Dogs receiving target therapy had a median time to progression of 363 days, while dogs treated with metronomic chemotherapy had only 27 days (*p* = 0.079) [73]. Sorafenib could be a promising targeted therapy for canine HSA [72,73,90]; however, its cost may be a barrier for many pet owners.

The RAS-RAF-MEK-ERK pathway, also known as MAPK, consists of a signaling cascade for cell proliferation and differentiation [80,82,90]. RAS and RAF oncogenes have attracted the most attention within this group due to their well-described carcinogenic potential [79,81,82,83,85,86]. Nonetheless, there are many other genes and proteins of potential importance, including the mitogen-activated protein/extracellular signal-regulated kinase (MEK). Although not tested in canine HSA, MEK can be inhibited by trametinib or cobimetinib [90,91,92]. RAF functions by phosphorylating and activating MEK, which, subsequently, phosphorylates and activates ERK, ultimately resulting in gene transcription [85] (Figure 3).

The in vitro inhibition of MAPK has been demonstrated to downregulate tumoral growth in canine HSA cell cultures of primary splenic, cardiac and cutaneous presentations [90]. Three MEK inhibitors, CI-1040, Sorafenib and LY294002, were tested and provided evidence that, even in vivo, the inhibition of this pathway is able to suppress both human angiosarcoma and canine HSA proliferation [90].

### 3.2. PIK3CA

The phosphatidylinositol 3-kinase (PI3K) is an oncogene product, featured by a catalytic unit and considered a potential anticancer target. The phosphatidylinositol-4,5-bisphosphate 3-kinase catalytic subunit alpha (PIK3CA) plays a fundamental role, acting in the AKT-mTOR pathway, encoding the P110α oncoprotein and regulating this cellular metabolic pathway in several human cancers [51,93,94,95]. The PI3K-AKT-mTOR is considered one of the most important pathways in many cancers including vascular tumors. The independent phosphorylation of AKT (Ser473) was associated with cell cultures of canine HSA in the absence of PTEN deletion and unaltered by fetal bovine serum, suggesting a constitutive activation of the PI3K/Akt/mTOR pathway in these cell lines [95]. Dysregulation of the PI3K-AKT-mTOR pathway leads to increased proliferation signaling and dysregulation of cellular metabolism—a hallmark of cancer [94,96,97].

Mutations in PIK3CA are mostly identified in coding domains of P110α, a major hotspot, although different domains might be mutated according to the origin of the cancer [94]. The modification of amino acid 1047, responsible for P110α protein loop conformation, seems to be frequently mutated in canine HSA cases, representing 71% (10/14) of all mutations in PIK3CA and characterized as H1047R or H1047L [40,44,61].

In both human and veterinary medicine, the mutation of PI3K in general has a negative impact on survival time independently of the cancer type [42]. Targeting PIK3 appears to be an interesting anticancer treatment strategy, and in human medicine, Alpelisib and Copanlisib, two PIK3 inhibitors, have been approved for the treatment of various cancers [78]. A recent study assessed the impact of Alpelisib on PIK3CA-mutated canine HSA cell lines [61]. This PI3K inhibitor exhibits potential as a targeted therapy for dogs with PIK3CA mutations: in vitro effects demonstrated its capacity to inhibit cancer cell migration, suppress AKT phosphorylation, thereby inhibiting cell proliferation, and induce apoptosis through caspase-3/7 activation [61]. Clinical trials could provide further evidence regarding the effectiveness of Alpelisib for canine HSA and the potential of PIK3CA mutations as predictive markers for this therapy.

Gedatolisib is a drug under development used in cancer patients with mutations in PIK3CA. It inhibits the P110α catalytic subunit of the PI3K gene, inhibiting the PI3K/mTOR pathway [75,76,77,98]. Recent evidence shows that it declines the viability of canine tumor cells of specific cell lines, such as osteosarcoma and histiocytic sarcoma [75]. However, trials with PI3K in HSA and angiosarcoma in canine and human patients are lacking. Considering the high percentage of PI3K mutation in canine HSA and the preliminary in vitro results, the inhibition of PI3K could be a viable therapeutic strategy for this type of tumor [99].

The occurrence of a specific mutation in an oncogene does not exclude the possibility of mutations in other oncogenes; a synergism effect may occur in tumors with additional mutations, such as in different segments of the PIK3CA-AKT-mTOR pathway and PIK3CA mutations coexisting with different PI3K mutations in endometrial, breast and colorectal human cancer [100]. Coexisting mutations in PIK3CA and TP53 have been occasionally identified in splenic and cardiac canine HSA [44].

### 3.3. PTEN

The PTEN is a tumor-suppressor gene, located in chromosome 10, responsible for the regulation of the PI3K-AKT-mTOR pathway through the inhibition of PI3K/Akt signaling [51,101,102,103,104]. Like TP53, PTEN is frequently mutated in animal neoplasms, and a loss of PTEN upregulates the PI3K-AKT-mTOR pathway, facilitating cell cycle progression and increasing proliferation and anabolic metabolism throughout enhancing protein synthesis in cancer cells [104]. PTEN mutations are frequently found in breast and prostate cancer in humans, where there is a positive correlation between the loss of PTEN and adverse outcomes and poor prognosis [51,103]. Mammary gland tumors might also be associated with dysregulations of PTEN expression [101,105]. In addition, there is evidence that PTEN may regulate more aspects of the cancer’s behavior independently of the PI3K-AKT-mTOR pathway, affecting the tumoral microenvironment and immunomodulation that may affect the clinical response to immunotherapy [50,51,104].

Reported almost two decades ago [64], PTEN mutations in canine HSA can occur in 4% (2/47) to 10% (2/20) of cases [40,41,43,44] and appear to play an important role in regulating PIK3CA, a more frequently mutated gene in this tumor [41,44]. PTEN is also downregulated in human angiosarcomas of the scalp and face, documented by reduced expression in immunohistochemistry for malignant and higher-grade tumors [49]. Inactivation of PTEN leads to upregulation of the mTOR pathway and targeting the mTOR pathway could be an effective anticancer strategy [102].

Specific mTOR inhibitors, Temsirolimus and Everolimus, have been successfully used and approved for the treatment of renal cell carcinoma in people [106,107]. Despite the potential efficacy in vascular tumors, mTOR inhibitors have not been investigated in clinical trials in angiosarcoma in people. However, Everolimus showed some modest efficacy in people with bone and soft tissue sarcomas [108].

For canine mammary neoplasms, both Temsirolimus and Everolmius have exhibited evidence to inhibit the mTOR pathway in vitro, leading to suppressed growth of tumor-adherent cells and spheres [109]. Additionally, Everolimus presents an anti-growth potential in sphere-formation assays related to cancer stem cells with self-renewal ability. However, the precise antitumor effect of these inhibitors in this model remains uncertain, as no discernible differences were observed between the control group and the groups treated with mTOR inhibitors regarding mitotic figures count and angiogenesis [109]. Sirolimus (rapamycin), the first discovered mTOR inhibitor, has not been investigated in dogs with HSA. However, adding Sirolimus to the standard of treatment for canine appendicular osteosarcoma has not shown significant benefit in overall survival, compared to the standard of treatment alone [110]. The efficacy of Sirolimus in combination with the standard treatment of care for hemangiosarcoma is under investigation, but it is unlikely to be effective alone in a disease with a rapid course and aggressive behavior. A combination of various treatments, including standard chemotherapy, mTOR inhibitors and repurposed drugs such as β-blockers, could be more effective [111].

### 3.4. TP53

This tumor-suppressor gene is commonly affected in neoplasms from different origins in humans and domestic animals [41]. It is the gene most frequently mutated in canine HSA, thus playing a meaningful role in its growth mechanism, most importantly avoiding tumoral suppression when mutated [43,64,65]. TP53 is also able to promote the fusion of genes through genomic instability [44]. In immunohistochemistry, cutaneous HSA with actinic damage expressed a higher p53 index when compared to visceral manifestations and cutaneous forms without actinic changes [26]. In a recent study, mutations of TP53 were shown to be a negative prognostic factor, independently of cancer type. Furthermore, the most common cancer with this mutated tumor-suppressor gene was canine HSA [42]. Alsaihati et al. (2021) identified a similar pattern, in which HSA was the tumor with the highest frequency of TP53 mutations (59%) among the analyzed neoplasms, especially in Golden Retrievers, although a prognostic analysis was not elaborated [41].

Therapies targeting p53 could be very useful in preserving genomic stability and regulating the cell cycle in cancers [112,113,114]. Gene therapy works by targeting specific mutations and restoring the functionality of the affected gene [67,113]. Gene therapy restoring the functionality of p53 can be theoretically used in a wide variety of cancer types; however, the efficacy of this treatment is largely unknown for many tumors with a mutation of p53 [113]. The recombinant human p53 adenovirus (Gendicine) was the first gene therapy to be approved for the treatment of head and neck carcinomas with p53 loss in humans. Gendicine increases response rate and overall survival time when associated with chemotherapy and/or radiotherapy in head and neck carcinoma [66,67]. Unfortunately, p53 genetic therapy is not yet available in small animal oncology and no studies have evaluated the efficacy of this treatment in dogs with cancer.

### 3.5. CDKN2A

The cyclin-dependent kinase inhibitor 2A (CDKN2A) is a tumor-suppressor gene that controls the cell cycle, mostly in the G1 to S phase, by encoding proteins p16 and p14 [115,116]. Aberrations in this gene lead to tumorigenesis and metastasis as a result of a lack of cell cycle regulation [116]. In humans, it has been associated with various tumors, including pancreatic cancer, melanoma and angiosarcoma [117,118,119]. An experimental study tested the inoculation of oncogenic HRAS with the knockdown of CDKN2A or TP53 in mice and detected that these associations were able to induce the development of angiosarcomas and other soft tissue sarcomas [119]. In canine HSA, few studies analyzed CDKN2A; nonetheless, recently it was detected that 11% of dogs with HSA expressed mutations in this gene [40].

### 3.6. VEGF, Angiogenesis and Hypoxia

Vascular endothelial growth factor (VEGF) is a signal protein that induces the development of blood vessels, both in physiological and pathological conditions [120]. Its dysregulation enhances cancer growth by overstimulating angiogenesis, already supported by tumoral chemokines. As a result, the angiogenesis leads cancer cells to proliferate despite hypoxia [120,121]. Different types of VEGF play distinct functions in blood and lymphatic vasculature development, even in tumoral conditions, most notably VEGF-A and VEGF-D, usually overexpressed in breast cancer and angiosarcoma in humans [122,123,124]. Hypoxia is a dichotomous factor in some neoplasms [120,124,125]. Despite the need for nutrients and blood supply for a tumor cell to survive, some cancers, such as angiosarcoma, may have better development in a hypoxic microenvironment, evading immune antitumoral response and facilitating proliferation and migration [121,125,126,127]. The lack of tumoral oxygenation and the overexpression of VEGF may be poor prognostic factors that alter the cancer microenvironment in order to promote growth [123,125,128].

Vascular neoplasms commonly express VEGF and its receptors, and its overexpression appears to accelerate tumor proliferation [124,129,130]. The low expression of circulating VEGF-A in people with vascular sarcomas treated with sorafenib, including angiosarcoma, is correlated with a more favorable outcome [48]. A similar pattern occurs with low serum rates of VEGF-C, which are associated with poor prognosis and shorter disease-free time in humans with angiosarcoma treated with paclitaxel and bevacizumab [131]. No studies reported correlations between tyrosine kinase inhibitors or commercial monoclonal antibodies and VEGF expression or its blood rate in dogs with any type of cancer.

In canine cutaneous HSA, no correlation has been identified between VEGF expression and overall survival time [26]. Splenic HSA in dogs expressed a higher number of VEGF+ cells in comparison to splenic hemangiomas (*p* = 0.004) [132] and an average staining expression of VEGFR-2 that was four times higher than in normal spleens [130]. Evidence suggests that VEGF and VEGFR-2 may be useful diagnostic markers, although neither VEGF nor its receptors are well established as predictive or prognostic factors for canine HSA yet [26,130,132]. Antibodies that bind VEGF, VEGFR blockers and modified receptors are therapeutic options potentially able to reduce VEGF action in angiogenesis in canine cancer patients. Sorafenib is a tyrosine kinase that inhibits VEGFR-2 and PDGFR, suitable for human angiosarcoma treatment [48,133] and well tolerated by dogs [72,74]. Thus, it could be effective in disease control as a canine-HSA-targeted therapy, considering the high expression of VEGFR-2 in splenic HSA cells [130]. However, the use of toceranib, a TKI that blocks VEGFR, has not proven to be beneficial in splenic HSA after the standard treatment [29]. Immunotherapy is not frequently used for vascular tumors. Bevacizumab (Avastin^®^; Genentech, Inc., South San Francisco, CA, USA) is a monoclonal antibody anti-VEGF, potentially inhibiting tumoral angiogenesis, considered as a promising therapy, but in recent studies was found to not be beneficial for human angiosarcoma treatment [134].

### 3.7. PD-1/PD-L1 Complex

Immune checkpoint inhibitors (ICI) and vaccines are promising treatment modalities for several tumors in humans and domestic animals. Studies in the last decade have shown some evidence of efficacy for treating invasive urothelial carcinoma and melanoma in dogs, namely Oncotherad nano-immunotherapy and the Oncept vaccine [135,136,137,138,139]. Few studies have focused on evaluating the PD-1/PD-L1 complex as a potential target for canine HSA [70,140,141]. Programmed cell death protein 1 (PD-1) is a lymphocyte surface protein that, when bound to its ligand, programmed death-ligand 1 (PD-L1), can downregulate T cells. This interaction acts as a tumor immune system evasion mechanism [70]. The pathways of this mechanism include the inhibition and apoptosis of tumor-infiltrating T lymphocytes, reduced secretion of pro-inflammatory cytokines and cell cycle arrest in the G0/G1 phase [70,141,142].

The ISOS-1 (RRID:CVCL_C517) angiosarcoma cell line shares molecular similarities with canine hemangiosarcoma (HSA) [140]. Notably, a study revealed that these tumor cells exert control over macrophages, polarizing them into the M2 phase, associated with pro-tumor effects, and upregulating the expression of PD-L1. This ligand expression is observed in most canine HSA cases [140], and also in tumor-infiltrating macrophages, indicating a pro-tumor role of the PD-1/PD-L1 complex in canine HSA [140].

The in vivo effectiveness of an anti-PD-1 immune checkpoint inhibitor (ca-4F12-E6) was documented in two dogs diagnosed with melanoma [68]. These patients exhibited a complete response that lasted over a year, surpassing the expected outcomes of conventional therapy [68,71]. Additional studies are warranted to evaluate the therapy’s efficacy and safety, involving a larger cohort of dogs and comprehensive safety assessments. Currently, some effective antibodies targeting PD-1 and PD-L1, such as Pembrolizumab and Atezolizumab, have been approved for the treatment of human tumors [69,70]. However, in veterinary medicine, there are no commercially available immune checkpoint inhibitors yet.

## 4. Conclusions and Future Directions

The prognosis for visceral canine HSA remains poor. Recent studies have improved our knowledge of the most common molecular profile of HSA in dogs, clarifying different patterns according to each presentation of the neoplasm.

In our study, the most frequently analyzed mutated genes in canine HSA identified in recent articles are TP53, PIK3CA and NRAS. Nonetheless, few investigations in recent decades have searched for mutations in other genes, such as CDKN2A, PTEN and AKT1, which could provide more knowledge regarding this tumor’s resistance and etiopathogenesis. Targeting one, or more likely several dysregulated molecular pathways, such as RAS-RAF-MEK and AKT-mTOR, could be beneficial in treating this fatal disease. Thus, new studies should test different strategies focused on specific pathways.

As demonstrated previously, canine HSA’s molecular profile differs from other neoplasms. A different molecular background could even be present in different portions of the same neoplasm. In the near future, the treatment of HSA should focus on a more personalized treatment, based on the specific molecular profile of individual dogs with HSA.

## Figures and Tables

**Figure 1 vetsci-10-00387-f001:**
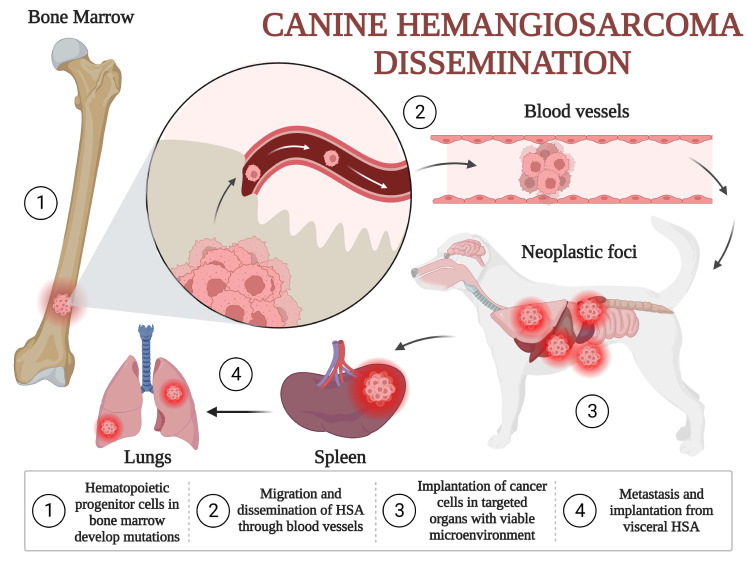
Canine hemangiosarcoma dissemination: (1) Hematopoietic progenitor cells in bone marrow develop mutations and from non-neoplastic stem cells transform to HSA cells. (2) Initial migration and dissemination of the tumor occur from the bone marrow to systemic circulation, able to reach multiple organs. (3) Cancer cells reach and colonize targeted organs with viable microenvironments, such as the spleen, liver and right atrium. (4) Secondary dissemination may occur from the tumors to lungs or through cavitary implantation in cases of rupture, as in the peritoneum and omentum [15,16,19,20].

**Figure 2 vetsci-10-00387-f002:**
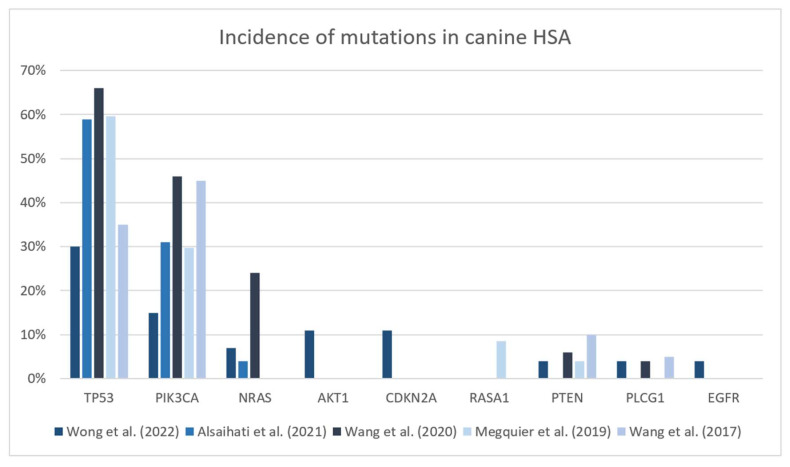
Incidence of mutations in cancer-related genes in canine hemangiosarcoma cases [40,41,43,44,53].

**Figure 3 vetsci-10-00387-f003:**
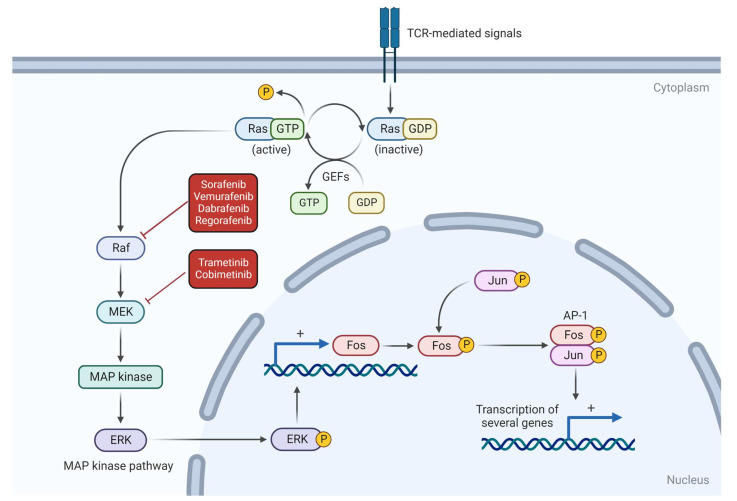
RAS-RAF-MEK pathway and potential therapeutic targets. RAS protein leads to RAF activation, which can be inhibited by sorafenib, vemurafenib, dabrafenib and regorafenib. MEK protein, activated by RAF, can be inhibited by trametinib and cobimetinib.

**Table 1 vetsci-10-00387-t001:** Genes associated with etiopathogenesis of canine HSA [40,41,42,43,44,47,53,56,63,64,65].

Gene	Role	Reference
AKT1	Oncogene	Wong et al. (2022) [40].
CDKN2A	Tumoral supressor gene	Wong et al. (2022), Thomas et al. (2014) [40,47].
EGFR	Oncogene	Wu et al. (2023), Wong et al. (2022) [40,42].
NRAS	Oncogene	Wu et al. (2023), Wong et al. (2022), Alsaihati et al. (2021), Kim et al. (2021), Wang et al. (2020) [40,41,42,43,44].
PIK3CA	Oncogene	Wu et al. (2023), Wong et al. (2022), Alsaihati et al. (2021), Kim et al. (2021), Wang et al. (2020), Megquier et al. (2019), Wang et al. (2017) [40,41,42,43,44,53,65].
PTEN	Tumoral supressor gene	Wong et al. (2022), Wang et al. (2020), Megquier et al. (2019), Wang et al. (2017), Dickerson et al. (2005) [40,43,53,63,65].
RASA1	Oncogene	Wong et al. (2021), Megquier et al. (2019) [64,65].
TP53	Tumoral supressor gene	Wu et al. (2023), Wong et al. (2022), Alsaihati et al. (2021), Kim et al. (2021), García-Iglesias et al. (2020), Wang et al. (2020), Megquier et al. (2019), Wang et al. (2017) [40,41,42,44,53,56,65].

**Table 2 vetsci-10-00387-t002:** Targets and potential targeted therapies for canine HSA [45,46,61,66,67,68,69,70,71,72,73,74,75,76,77,78].

Targets	Targeted Therapies, Experiment Conditions (Species)	Reference
Histone acetylation	JQ1/BETi, in vitro and in vivo (humans and dogs).	Neganova et al. (2022), Suzuki et al. (2022) [45,46].
p53	Gendicine/recombinant human p53 adenovirus, in vitro and in vivo (humans).	Hasbullah and Musa (2021), Zhang et al. (2018) [66,67].
PD-1/PD-L1 complex	Pembrolizumab/ICI, in vitro and in vivo (humans). Atezolizumab/ICI, in vitro and in vivo (humans) and ca-4F12-E6 (dogs).	Igase et al. (2022), Pantelyushi et al. (2021), Jiang et al. (2019) [68,69,70,71].
PDGFR	Sorafenib/TKI, in vitro and in vivo (humans and dogs).	Cawley et al. (2022), Marconato et al. (2020), Foskett et al. (2017) [72,73,74].
PI3K/AKT/mTOR	Alpelisib/PIK3 inhibitor, in vitro (humans and dogs) and in vivo (humans), and Gedatolisib/PIK3 inhibitor, in vitro and in vivo (humans).	Maeda et al. (2022), Murase et al. (2022), Wilson et al. (2021), Liu et al. (2021), Markham (2019) [61,75,76,77,78].
RAS-RAF-MEK	Sorafenib/TKI, in vitro and in vivo (humans, dogs).	Cawley et al. (2022), Marconato et al. (2020), Foskett et al. (2017) [72,73,74].
VEGFR	Sorafenib and Toceranib/TKI, in vitro and in vivo (humans and dogs).	Cawley et al. (2022), Marconato et al. (2020), Foskett et al. (2017) [72,73,74].

BETi: bromodomain and extraterminal domain inhibitor; ICI: immune checkpoint inhibitor; TKI: tyrosine-kinase inhibitor.

## Data Availability

The data analyzed within this article are considered secondary due to the study type (narrative review). Any inquiries regarding our data should be directed to the designated corresponding author.

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
