# Peer review of "Molecular Profile of Canine Hemangiosarcoma and Potential Novel Therapeutic Targets"

_vetsci, 2023, doi:10.3390/vetsci10060387_

Round 1

Reviewer 1 Report

Reviewer comment

The authors summarize what is known to date about the relationship between genetic mutations that occur in canine hemangiosarcoma and the pathogenesis of the disease, and describe a review that may aid in the development of new therapeutic strategies in this paper (vetsci-2424072). The article focuses on the function of various cancer-related genes found to be mutated in canine angiosarcoma, such as PIK3CA, NRAS, TP53, PTEN and CDKN2A, and their relationship to the pathogenesis of the disease, making it an extremely valuable review. The paper is detailed and carefully written and is acceptable as is, but I believe the quality could be further improved by adding the following minor points.

Lines 222-224: Please mention a paper examining the antitumor effect of Alpelisib on a canine hemangiosarcoma-derived cell line in which residue 1047 of PIK3CA is mutated. (Maeda et al., 2022, Oncol Rep. 47(4):84).

Various papers on canine hemangiosarcoma cell lines have been published and should be cited more actively. (e.g. Murai et al., BMC Vet Res. 2012, 29(8):128.; Igase et al., Exp Cell Res. 2020 388(1):111810.)

Lines 262-263: Specific mTOR inhibitors, Temsirolimus and Everolimus, have been reported to have growth inhibitory effects on cancer stem cells derived from a canine mammary tumor cell line, and I recommend that these be added as examples of use in dogs. (Michishita et al., Front Oncol. 2023, 13:1100602.)

Author Response

Response to Review 1:

We would like to thank the reviewer for these comments and useful suggestions. Each suggestion and comment were addressed below according to its respective response, point by point. In the text, every change was marked up using the “Track Changes” function. Hope this message finds you well.

Point 1: Lines 222-224: Please mention a paper examining the antitumor effect of Alpelisib on a canine hemangiosarcoma-derived cell line in which residue 1047 of PIK3CA is mutated. (Maeda et al., 2022, Oncol Rep. 47(4):84).

Response 1: This paper is exceptionally interesting and valuable as it sheds light on a potential therapy and perhaps a predictive marker. It also improves our comments in the section of PIK3CA mutations. The article has been duly referenced, commented on, and a citation was included.

Point 2: Various papers on canine hemangiosarcoma cell lines have been published and should be cited more actively. (e.g. Murai et al., BMC Vet Res. 2012, 29(8):128.; Igase et al., Exp Cell Res. 2020 388(1):111810.)

Response 2: We agree with the suggestion. HSA cell line studies are now more cited throughout our study. These studies were included and cited in its respective areas. We have also included other studies regarding this topic. The following studies were included: Murai et al. (2012); Igase et al. (2020); Maeda et al. (2022); Kim et al. (2014); Im et al. (2017).

Point 3: Lines 262-263: Specific mTOR inhibitors, Temsirolimus and Everolimus, have been reported to have growth inhibitory effects on cancer stem cells derived from a canine mammary tumor cell line, and I recommend that these be added as examples of use in dogs. (Michishita et al., Front Oncol. 2023, 13:1100602.)

Response 3: We agree with the recommendation. Still there is few evidence of these inhibitors for dogs and no evidence for canine HSA. Although evidence has been found indicating that Temsirolimus and Everolimus possess the ability to inhibit the mTOR pathway. Particularly, we think that Everolimus should be tested more actively. This suggestion was now referenced, included and discussed in mTOR inhibitors (PTEN section).

Additional alterations:

  1. Figure 3 (RAS-RAF-MEK pathway and potential therapeutic targets) added in high resolution.
  2. Fixes in inadequate Word® automatic modifications (e.g. HSA substituted to HAS) and duplicated references.
  3. As requested by the editor to expand number of figures/tables, a new figure was added (Canine hemangiosarcoma dissemination, Figure 1) and a new table was included (Targets and potential targeted therapies for canine HSA).

Reviewer 2 Report

The manuscript reviews some of the recent knowledge gathered on the most frequent mutations present in canine hemangiosarcoma, and their impact on new therapeutic solutions.

There are other recent works addressing the subject through a One health perspective, namely the ones by Megquier et al. (2019), duly cited in the present work.

In the introduction, the authors refer to diagnosis through liquid biopsy as described by Flory et al. (2022) and it would be interesting to discuss the measurement of plasma nucleosome concentrations as a screening test in hemangiosarcoma, as described by Wilson-Robles et al. (2021).

The present review may be improved by including some works and aspects of angiosarcoma and canine hemangiosarcoma that are worth pondering and discussing, namely those on PD-1 and PD-L1 expression, present in approximately 60% of canine hemangiosarcoma samples in the work by Maekawa et al. (2016), and its influence in the immune cells such as macrophages (Gulay et al., 2022) and lymphocytes and on the use of immune checkpoint blockers, as these are beneficial in humans, in the treatment of angiosarcomas and other aggressive soft tissue sarcomas.  The functionality and cross-reactivity have been evaluated by Pantelyushin et al. (2021) and Igase et al. (2022), for example.

Discussion on the changes in histone acetylation pathways and its potential therapeutic implications should also be included, and proper references added.

These changes in the manuscript should be reflected in Table 1. It would be interesting to add a table summarizing the recent studies published on in vitro and in vivo evaluation of these new therapeutic approaches.

The manuscript is well-written and informative. In line 338 I suggest changing "well-tolerated for dogs" to "well-tolerated by dogs".

Author Response

Response to Review 2:

We would like to thank the reviewer for these comments and useful suggestions. Each suggestion and comment were addressed below according to its respective response, point by point. In the text, every change was marked up using the “Track Changes” function. Hope this message finds you well.

Point 1: In the introduction, the authors refer to diagnosis through liquid biopsy as described by Flory et al. (2022) and it would be interesting to discuss the measurement of plasma nucleosome concentrations as a screening test in hemangiosarcoma, as described by Wilson-Robles et al. (2021).

Response 1: We fully concur with this suggestion. Liquid biopsies hold tremendous potential as the future of non-invasive tumor diagnosis. We have cited, referenced, and included the study supporting this notion. The concept of liquid biopsies as a non-invasive screening test for diagnosis is both fascinating and highly promising.

Point 2. The present review may be improved by including some works and aspects of angiosarcoma and canine hemangiosarcoma that are worth pondering and discussing, namely those on PD-1 and PD-L1 expression, present in approximately 60% of canine hemangiosarcoma samples in the work by Maekawa et al. (2016), and its influence in the immune cells such as macrophages (Gulay et al., 2022) and lymphocytes and on the use of immune checkpoint blockers, as these are beneficial in humans, in the treatment of angiosarcomas and other aggressive soft tissue sarcomas.  The functionality and cross-reactivity have been evaluated by Pantelyushin et al. (2021) and Igase et al. (2022), for example.

Response 2: We sincerely appreciate and fully agree with this recommendation. In line with the suggestion to incorporate these works, we have made the decision to add a new section specifically dedicated to PD-1/PD-L1. This section provides a discussion on the PD-1/PD-L1 complex and the current evidence pertaining to tumors in dogs, including canine HSA. We have ensured that all the requested studies, namely Maekawa et al. (2016), Gulay et al. (2022), Pantelyushin et al. (2021), and Igase et al. (2023), have been appropriately cited, referenced, and included in our work.    

Point 3: Discussion on the changes in histone acetylation pathways and its potential therapeutic implications should also be included, and proper references added.

Response 3: We acknowledge and agree with the notion that histone acetylation may hold significant relevance in the etiopathogenesis of canine Hemangiosarcoma (HSA). However, the precise role it plays in HSA cells remains unclear. In light of this, we have provided a concise yet informative discussion on this topic.

Point 4: These changes in the manuscript should be reflected in Table 1. It would be interesting to add a table summarizing the recent studies published on in vitro and in vivo evaluation of these new therapeutic approaches.

Response 4: We have allocated Table 1 specifically for mutations, and we have now introduced a new Table dedicated to presenting the identified therapeutic targets. This arrangement allows for a clear and organized presentation of the respective information.

Point 5: The manuscript is well-written and informative. In line 338 I suggest changing "well-tolerated for dogs" to "well-tolerated by dogs".

Response 5: Thank you for your suggestion. We apologize for the mistake.

Additional alterations:

  1. Figure 3 (RAS-RAF-MEK pathway and potential therapeutic targets) added in high resolution.
  2. Fixes in inadequate Word® automatic modifications (e.g. HSA substituted to HAS) and duplicated references.
  3. As requested by the editor to expand number of figures/tables, a new figure was added (Canine hemangiosarcoma dissemination, Figure 1) and a new table was included (Targets and potential targeted therapies for canine HSA).

Reviewer 3 Report

This is a comprehensive summary of literature regardingcell origin, anatomical sites, diagnosis, mutations and therapeutical approaches in canine hemangiosarcomas. Well done.

Otherwise I am missing critical appraisal in assessing the quality of research you collected. What was the recruitment strategy when collecting the information? Which databases did you use to collect the literature? Which information did you exclude and why? Please elaborate this and insert a figure documenting number of articles collected and excluded.

The quality of English language is fine

Author Response

Response to Review 3:

We would like to thank the reviewer for the comments and useful suggestions. In the text, every change was marked up using the “Track Changes” function. Hope this message finds you well.

Point 1: Otherwise I am missing critical appraisal in assessing the quality of research you collected. What was the recruitment strategy when collecting the information? Which databases did you use to collect the literature? Which information did you exclude and why? Please elaborate this and insert a figure documenting number of articles collected and excluded.

Response 1: Our primary aim was to provide an elaborate narrative review. However, conducting a comprehensive assessment of the specific number of collected and excluded articles would necessitate the development of a systematic review from scratch, including the establishment of inclusion and exclusion criteria. Given the diverse nature of the data analyzed in our study and its recent presence in the literature (for instance, some mutations have only been evaluated in a single article thus far, and most potential targets are still under evaluation), it would be challenging to draw systematic conclusions considering the broad scope of the topic. Consequently, we have opted for a narrative perspective.

Nonetheless, in accordance with the suggestion provided, we have included a paragraph outlining the process of data collection, even though we do not possess the precise figures regarding the number of studies selected and excluded.

Additional alterations:

  1. Figure 3 (RAS-RAF-MEK pathway and potential therapeutic targets) added in high resolution.
  2. Fixes in inadequate Word® automatic modifications (e.g. HSA substituted to HAS) and duplicated references.
  3. As requested by the editor to expand number of figures/tables, a new figure was added (Canine hemangiosarcoma dissemination, Figure 1) and a new table was included (Targets and potential targeted therapies for canine HSA).

Round 2

Reviewer 2 Report

I consider the authors have properly addressed all the issues previously signaled, resulting in an much improved manuscript.

The added information in the text and the tables is relevant, and the added references too.